# Storage and export of microbial biomass across the western Greenland Ice Sheet

T. D. L. Irvine-Fynn [1✉], A. Edwards [2], I. T. Stevens [1,3,4], A. C. Mitchell [1], P. Bunting [1], J. E. Box [5], K. A. Cameron [2,5,6], J. M. Cook [2,4,7], K. Naegeli [1,8], S. M. E. Rassner [2], J. C. Ryan [9], M. Stibal [10], C. J. Williamson [11] & A. Hubbard [12,13]

The Greenland Ice Sheet harbours a wealth of microbial life, yet the total biomass stored or exported from its surface to downstream environments is unconstrained. Here, we quantify microbial abundance and cellular biomass flux within the near-surface weathering crust photic zone of the western sector of the ice sheet. Using groundwater techniques, we demonstrate that interstitial water flow is slow ($\sim 10^{-2}$ m d$^{-1}$), while flow cytometry enumeration reveals this pathway delivers $5 \times 10^8$ cells m$^{-2}$ d$^{-1}$ to supraglacial streams, equivalent to a carbon flux up to 250 g km$^{-2}$ d$^{-1}$. We infer that cellular carbon accumulation in the weathering crust exceeds fluvial export, promoting biomass sequestration, enhanced carbon cycling, and biological albedo reduction. We estimate that up to 37 kg km$^{-2}$ of cellular carbon is flushed from the weathering crust environment of the western Greenland Ice Sheet each summer, providing an appreciable flux to support heterotrophs and methanogenesis at the bed.

[1] Department of Geography and Earth Sciences, Aberystwyth University, Aberystwyth, UK. [2] Institute of Biological Environmental and Rural Sciences, Aberystwyth University, Aberystwyth, UK. [3] School of Geography, Politics and Sociology, Newcastle University, Newcastle-upon-Tyne, UK. [4] Department of Environmental Science, Aarhus University, Frederiksborgvej, Roskilde, Denmark. [5] Department of Glaciology and Climate, Geological Survey of Denmark and Greenland, Copenhagen, Denmark. [6] School of Geographical and Earth Sciences, University of Glasgow, Glasgow, UK. [7] Department of Geography, University of Sheffield, Sheffield, UK. [8] Institute of Geography and Oeschger Center for Climate Change Research, University of Bern, Bern, Switzerland. [9] Institute at Brown for Environment and Society, Brown University, Providence, RI, USA. [10] Department of Ecology, Faculty of Science, Charles University, Prague, Czechia. [11] Bristol Glaciology Centre, School of Geographical Sciences, University of Bristol, Bristol, UK. [12] Centre for Gas Hydrate, Environment and Climate, Department of Geosciences, UiT—The Arctic University of Norway, Tromsø, Norway. [13] Geography Research Unit, University of Oulu, Oulu, Finland. ✉email: tdi@aber.ac.uk

The Greenland Ice Sheet sequesters and exports organic carbon[1–4], yet many of the associated biogeochemical processes and pathways remain largely undocumented. One important carbon source is the ice sheet surface[5,6], which is host to diverse and active microbial assemblages[7–10]. Recent work has highlighted the global significance of these supraglacial microbial communities: they hold key roles in amplifying ice melt by lowering bare-ice albedo[9,11–14], and driving biogeochemical cycling of carbon[15,16], nitrogen[15,17] and anthropogenic contaminants[18,19].

During seasonal melt, across Greenland's $220 \times 10^3$ km$^2$ bare-ice ablation area[20], microbes, organic and inorganic debris and associated nutrients are washed downslope by the supraglacial meltwater drainage networks that develop[21]. These networks commonly terminate in moulins that provide meltwater pathways to the subglacial environment[22]. Such seasonal meltwater transfer affects the labile, bioavailable carbon exported from the ice sheet[3,4,23] and ultimately influences the microbial community composition and nutrient supply to downstream aquatic and marine ecosystems[24,25]. However, supraglacial drainage networks comprising streams, rivers and lakes account for a small fraction of the total ablating area of the ice sheet[26–28] and the majority of summer-season runoff is generated over extensive bare-ice areas between supraglacial meltwater channels[29]. The hydrological functioning of this wide-spread interfluvial area is unconstrained[28,30] and its role in modulating the delivery of microbes, organic and mineral dusts and debris, and associated nutrients to the subglacial environment remains unquantified.

The bare-ice interfluve area is characterised by the presence of an up to ~1 m thick, porous ice weathering crust, that forms due to subsurface shortwave radiation penetration, melting and percolation. In west Greenland, this weathered near-surface ice exhibits an effective porosity of up to 47% with a specific water storage potential of up to 0.18 m (ref. [31]) and has an assumed meltwater throughflow velocity of ~1–10 m d$^{-1}$ (ref. [28]). Smith et al.[32] demonstrate that the weathering crust has an important hydrological function and delays surface meltwater runoff, challenging the commonly held assumptions of efficient and rapid supraglacial drainage. Furthermore, the weathering crust photic zone is increasingly recognised as an important microbial habitat[33–35]; yet little is known about how, where and what quantities of microbial biomass are stored and produced within, or exported from, the weathering crust to downstream environments and ecosystems.

Here, we present the first quantitative assessment of microbe transport through the near-surface of the Greenland Ice Sheet and estimate the equivalent carbon fluxes exported from the weathering crust to the subglacial environment. In July 2014, at a site located on ice sheet's western margin (67° 04.78′N, 49° 24.08′W: Fig. 1a), we applied standard groundwater techniques to determine the conductivity of the near-surface ice[36] and employed flow cytometry to quantify the abundance of microbes entrained in the meltwater within the weathering crust. A first-order, catchment-scale model of microbial cell transport was developed by applying our observations across a high-resolution digital elevation model (DEM) derived from unmanned aerial vehicle (UAV) surveys (Fig. 1b). Our analysis reveals that within the overlooked supraglacial habitat of the weathering crust, in situ cellular accumulation exceeds microbial export; a finding that has significant implications for surface biogeochemical dynamics, carbon sequestration and cycling, and regional albedo reduction.

## Results and discussion
### Meteorology and meltwater production
The study period of July 23 to 29, 2014 was characterised by predominantly clear-sky conditions with a mean two-metre air temperature of 1.9 °C (Fig. 2) and consistent diurnal melt variability, typically peaking

at between 3 and 4 mm h$^{-1}$ at 13:00–14:00. Ninety local cryoconite holes and 57 shallow experimental auger holes indicated that the near-surface water table was located 7.5 (± 3.9) to 10.9 (± 5.4) cm below the ice surface, respectively. A total of 47 successful recharge experiments were conducted, yielding a mean hydraulic conductivity ($K$) of 0.28 (± 0.34) m d$^{-1}$ (Fig. 2d). The $K$-values derived from 23–26 cm and 34–36 cm deep auger holes are from statistically similar populations (U = 522.5, $p = 0.37$). Comparison of $K$ against melt rate (Fig. 3a) and daily melt cycle timing (Fig. 3b) indicates there is no strong interdependency, although water table height within the weathering crust correlates positively with $K$ ($\rho = 0.66$, $p < 0.001$) it is not associated with the instantaneous melt rate ($\rho = -0.12$, $p = 0.43$).

### Microbial abundance and mobility
The 73 water samples recovered from both fully and partially recharged auger holes in bare-ice show a mean microbial abundance of $2.28 \times 10^4$ cells mL$^{-1}$ (±$1.91 \times 10^4$ cells mL$^{-1}$ standard deviation) (Fig. 2e). Abundance exhibits no significant correlation with contemporaneous melt rate ($\rho = 0.07$, $p = 0.53$), but at the 95% confidence level suggests a slight negative relationship with time since daily peak melt ($\rho = -0.26$, $p = 0.03$), where both melt variables provide proxies for diurnal energy receipt and weathering crust development (see Fig. 3c, d). The cell size distribution from recharge waters (Fig. 4) reveals that the dominant size of SYBR Gold stained cells was 1–2 μm, representing 50% (± 7.0% standard deviation) of the suspended microbial abundance. The <1 μm category represents a mean of 19% (± 5.5%) of the total microbial abundance, but may include large viruses, thereby overestimating the true cell count. The microbial abundance in each of the six size classes were moderately to highly correlated ($0.28 < \rho < 0.93$, $p < 0.05$).

To examine the association between microbial abundance and ice surface hydrology, 26 completed recharge experiments were paired with coincident enumerations and reveal an inverse relationship between the hydraulic conductivity and microbial abundance, described by an exponential decay function (coefficient of determination $r^2 = 0.50$, $p < 0.001$: Fig. 5a). To differentiate between bacteria (and archaea) and larger algae, a 10 μm size classification threshold was applied[37]. Examination of the association between $K$ and abundance independently for these nominal bacteria and algae classes highlights similar non-linear, inverse relationships (respectively, $r^2 = 0.51$, $p < 0.001$ and $r^2 = 0.34$, $p < 0.002$: Fig. 5b), but with a reduced rate of algal abundance decline as the hydraulic conductivity increases.

### Microbial and carbon fluxes
To investigate the microbial fluxes within our supraglacial catchment, we assimilate our field-based measurements with a 1 m horizontal resolution DEM (Fig. 1b). Based on observations of the water table, we infer that porous ice extends to ~0.4 m depth with a hydrologically active, saturated weathering crust thickness of 0.29 m; by applying an effective porosity ($\phi$) of 22% (ref. [31]), the cross-sectional water flow area can be determined. Considering saturated transport through the weathering crust under Darcian flow[28,36], the throughflow velocity ($v_t$) is given by:

$$v_t = K\Delta h/\phi \tag{1}$$

where $\Delta h$ is the average local surface slope (0.022 m m$^{-1}$). With a mean value for hydraulic conductivity ($K$) of 0.28 m d$^{-1}$, we determine a spatially averaged weathering crust throughflow velocity of 0.028 m d$^{-1}$ and a specific meltwater discharge of 0.0018 m$^3$ d$^{-1}$. Typically, at our study site located within the western sector of the Greenland Ice Sheet, ablating bare-ice and, by inference, its hydraulically active weathering crust is exposed, on average, for around 70 days (or 76%) of the summer melt season[20,38]. Accordingly, using our derived mean and maximum $v_t$ values, respectively,

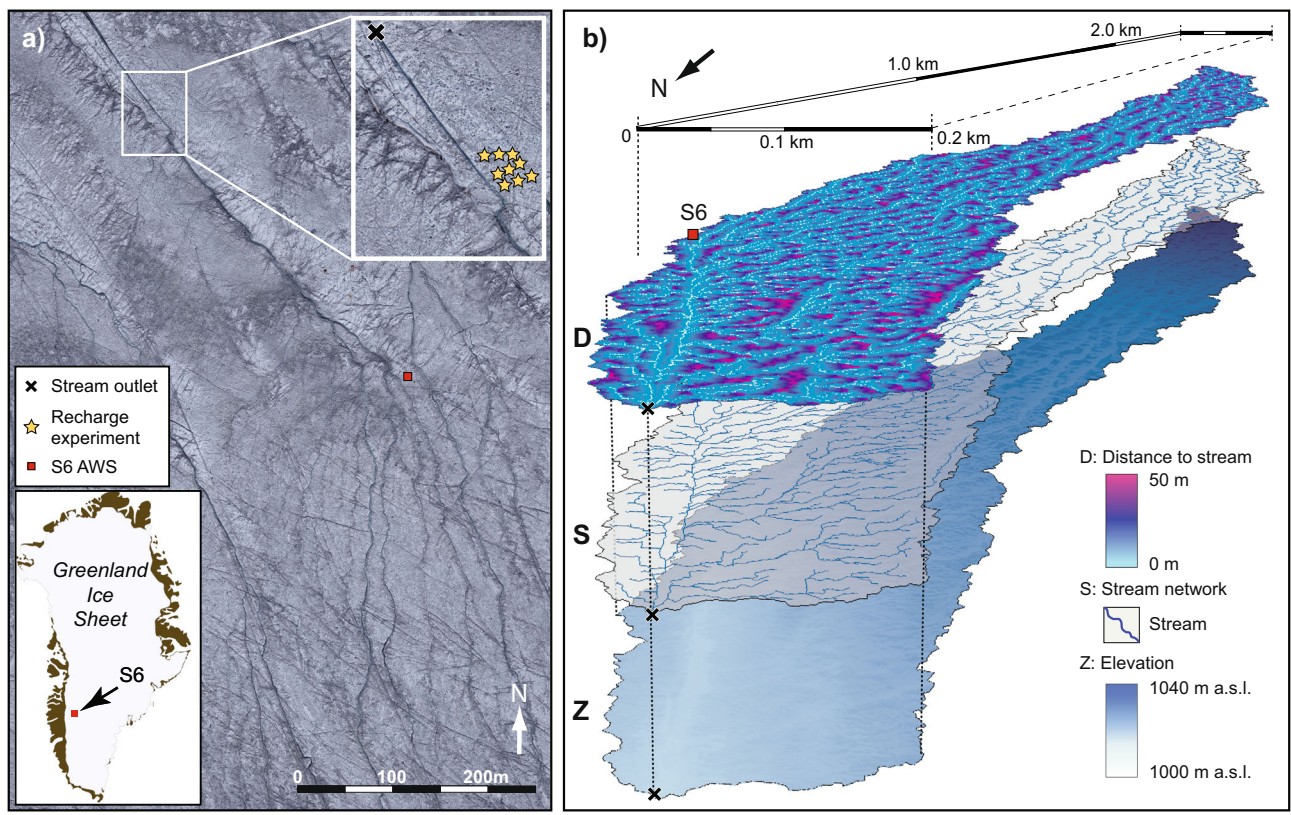

**Fig. 1 The S6 study site on the western margin of the Greenland Ice Sheet. a** High resolution unmanned aerial vehicle (UAV) derived digital red-green-blue (RGB) colour imagery of the ice sheet surface proximate to the S6 automatic weather station (AWS) to provide context to the bail-recharge experiments and experimental catchment outlet point (upper inset). **b** Staggered perspective maps of the study catchment topography (Z), the associated stream network (S), and flow-distance-to-stream metric (D).

only 22 to 66% of the catchment's weathering crust area delivers interstitial meltwater directly to the stream network seasonally (Fig. 1b).

From aerial photogrammetry of our study catchment, we calculate a bare-ice weathering crust area of $6.84 \times 10^4$ m$^2$ drained immediately (i.e., ≤1 m) into the stream network. Coupling this stream bank area with our estimates of mean throughflow velocity and mean microbial abundance we derive a daily microbial cell flux of $2.80 \times 10^{12}$, which equates to $5.73 \times 10^{12}$ cells km$^{-2}$ d$^{-1}$ when scaled across the entire catchment. Using our uppermost observed $K$-value, the specific discharge, total and specific cell fluxes increase to 0.008 m$^3$ d$^{-1}$, $1.27 \times 10^{13}$ cells d$^{-1}$ and $2.61 \times 10^{13}$ cells km$^{-2}$ d$^{-1}$.

To convert the daily cell fluxes to biomass estimates (see Methods), we apply a constant carbon content to derive a minimum, and a biovolume ratio to determine a maximum. Employing these approaches, for a conservative throughflow velocity, we propose the carbon efflux from the study catchment's interfluvial area (~0.32 km$^2$) to the supraglacial stream network lies between $8.36 \times 10^{-5}$ and 0.017 kg C km$^{-2}$ d$^{-1}$; on inclusion of the contribution from the larger (>15 μm) algal class, these values upwardly adjust to $1.36 \times 10^{-4}$ and 0.055 kg C km$^{-2}$ d$^{-1}$ (Table 1). This wide range of estimates highlights the dependence of carbon biomass approximations on the methods employed and the potential entrainment of larger cyanobacteria and algae.

**Weathering crust hydrology**. Our analysis demonstrates that an extensive saturated porous layer, with a water table extending from ~0.1 m below the surface, is hydraulically active in a study catchment located on the ablating western margin of the Greenland Ice

Sheet. Calculations of the weathering crust's hydraulic conductivity of $10^{-6}$–$10^{-5}$ m s$^{-1}$ and throughflow velocity of $10^{-7}$–$10^{-6}$ m s$^{-1}$ are comparable to those reported for unconsolidated, saturated silt and karstic sandstone[39], but are typically an order of magnitude lower than that reported for firn[40]. Our hydraulic conductivity compares well to other glacier surfaces across the northern hemisphere[36], but is somewhat lower than assessments made using different methods[41,42], and two orders of magnitude less than that theoretically estimated for Greenland by Yang et al.[28]. We propose that their coarse, 3 m resolution DEM and larger (>2000 m$^2$) stream-defining contributing area threshold aggregates the two hydrological functions of the slow interstitial near-surface matrix, and fast rills and micro-channels[43], yielding more rapid transit times. The hydrological function of the weathering crust is further compounded by macro-spatial variations in hydraulics, the saturated and unsaturated zones, near-surface fracturing processes and ice structure and crystal sizes[31,32,36,44]. Given these intrinsic environmental controls, the derivation of a spatially and/or temporally consistent estimate of the hydraulic conductivity from coarse resolution topographic metrics without any ground measurements is problematic.

Unsurprisingly, our throughflow velocities of <0.13 m d$^{-1}$ for the weathering crust are markedly lower than supraglacial stream and river velocities of $10^4$–$10^6$ m d$^{-1}$ (ref. [45,46]), but are similar to those reported for low-gradient ablating sea ice surfaces[47]. Though slow, the throughflow velocities could increase if the water table within the weather crust rose into the higher porosity uppermost unsaturated ice layer[31,44,48]. Our calculated meltwater efflux from the weathering crust to the stream network represents a relatively small proportion of the total discharge in the supraglacial channels, emphasising the potential importance of

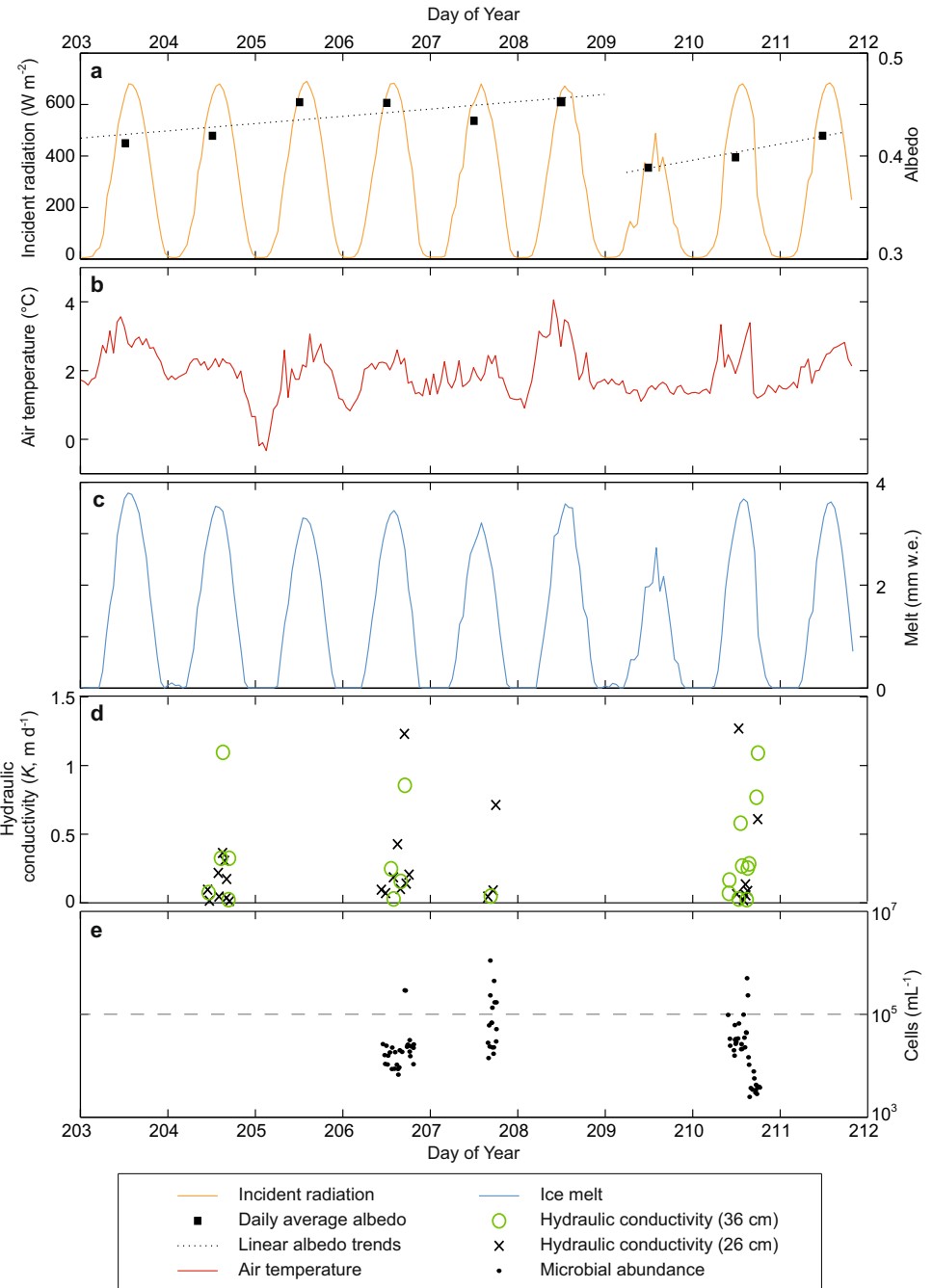

**Fig. 2 Time-series plots of hydrometeorological variables at the S6 weather station location during the study period in 2014. a** Record of incident shortwave radiation and daily mean surface albedo with associated temporal trends before and after the cloudy conditions on day of year (DOY) 209. **b** Air temperature over the 9-day study period. **c** Estimated ice melt according to a simple point-based energy balance model (see Methods). **d** The derived weathering crust hydraulic conductivity ($K$) values for 26 and 36 cm deep auger holes. **e** Microbial abundance in the recharge water samples associated with individual bail-recharge experiments; note the $10^5$ cells mL$^{-1}$ threshold (dashed line) used to define samples as outliers (see Methods).

flow contributions from within the unsaturated layer and micro-channels, and which merits further investigation. Nonetheless, the slow transit of water through the saturated weathering crust hydraulically acts to delay surface meltwater runoff[28,32] and impedes any associated microbe and nutrient transport.

**Microbial abundance and mobility.** Our estimates for micro-bial abundance in the recharge water compare well to other supraglacial enumerations using flow cytometry[10,49] as well as alternative methods[10,34,35,50,51]. We derive a microbial load of

$1.46 \times 10^9$ cells m$^{-2}$ for the hydraulically active weathering crust, with a minimum of $1.46 \times 10^8$ cells m$^{-2}$ (Table 1) taken as the background load. The three standard methods[52] applied to esti-mate net biomass stored in the near-surface yield a range of 0.02 to 4.3 kg C km$^{-2}$ excluding the largest algae, or 0.03 to 14.0 kg C km$^{-2}$ including these >15 μm cells. Assuming the allometric biomass method provides the most robust calculation[52], our best conservative estimate of the interfluvial weathering crust carbon load is 3.6 kg C km$^{-2}$, or 12.6 kg C km$^{-2}$ if algae are included. If we consider the inclusion of elevated microbial abundance related to discrete dust concentrations or

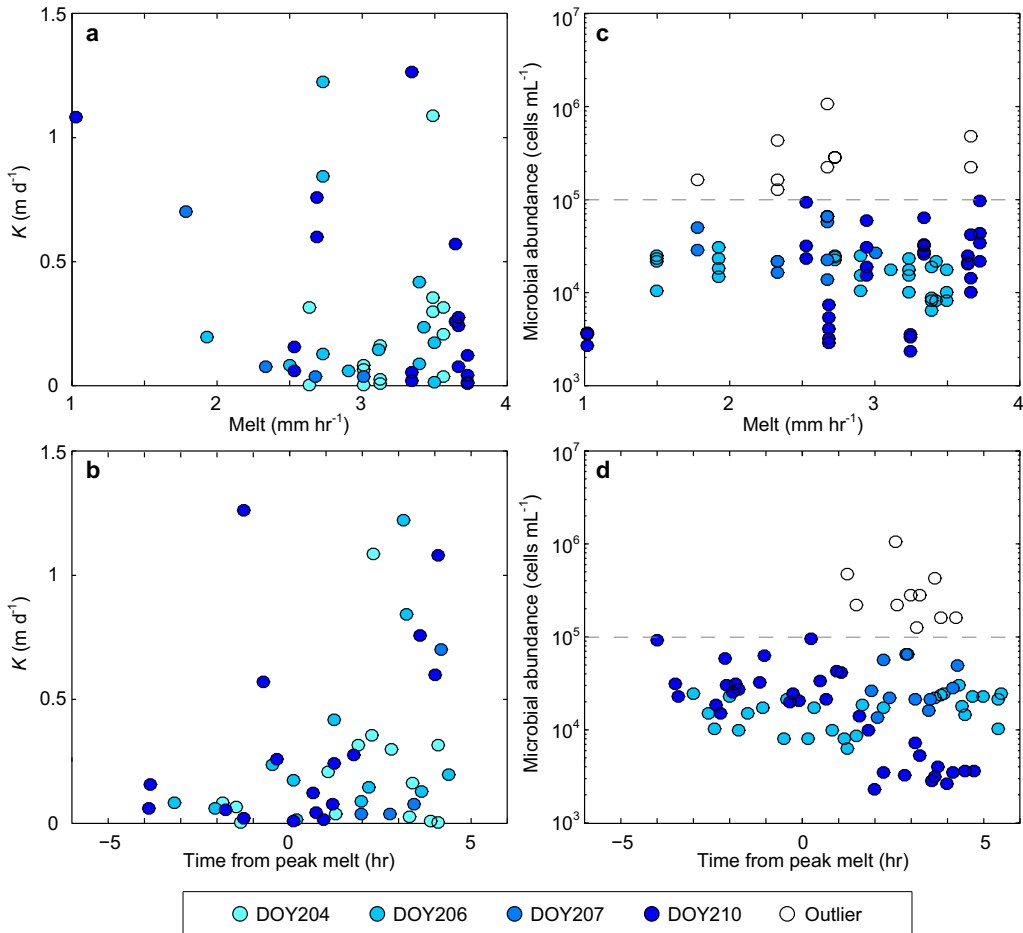

**Fig. 3 Scatter plots comparing melt conditions, near-surface hydraulic conductivity and microbial abundance. a** Relationship between hydraulic conductivity ($K$) and coincident melt. **b** Association between $K$ and time relative to peak melt. **c** Scatter plot of microbial abundance and coincident melt. **d** Scatter plot of microbial abundance and time relative to peak melt. Sample points are grouped and shaded according to the day of year (DOY) collection date, and those assessed as outliers, with $>1 \times 10^5$ cells $mL^{-1}$ (see Methods), are shown with hollowed markers above a dashed line.

algal blooms within our sample set (see Methods), the mean abundance is increased to $6.12 \times 10^4$ cells $mL^{-1}$, and our load rises to $3.9 \times 10^9$ cells $m^{-2}$ or 0.05 to 34.1 kg C $km^{-2}$.

The contrast in the hydraulic conductivity and abundance relationships between bacterial and algal categories suggests size-selective mobilisation: either bacteria are more rapidly depleted or excluded from transport as throughflow increases, or microbe availability is controlled by processes delivering or releasing cells from the ice surface or englacial environment. We suggest that these controlling processes may include mechanical filtering within the weathering crust ice matrix[49,53]; extrusion of exopolysaccharides (EPS) or other critical compounds[54–56]; or hydrological elution (flushing) with increased melt, analogous to the 'first flush effect'[57] at diurnal, synoptic and/or seasonal timescales as alluded to by the weak correlation between abundance and time since daily peak melt (Fig. 3d).

**Microbial export from western Greenland's weathering crust.** At our study catchment, we estimate that between $1.4 \times 10^{14}$ and $6.2 \times 10^{14}$ cells were liberated from the interfluve area to supraglacial stream transport during the 2014 ablation season, which had a bare-ice duration of 68 days. These values equate to 0.3 and 1.2 kg C (or 9.2 and 21 kg C if large algae are included) using an allometric best-estimate of biomass. If we consider our study catchment to be representative of all the moulin terminating supraglacial catchments across the ice sheet's western ablation

zone[22], upscaling our catchment data suggests that between $1.8 \times 10^{18}$ and $8.3 \times 10^{18}$ microbial cells were delivered to the subglacial drainage system over the 2014 ablation season. Accounting for the different biomass estimation methods, this equates to a seasonal carbon delivery of 25 kg C to $3.0 \times 10^5$ kg C from western Greenland's ablation zone under a mean throughflow velocity (or 42 kg C to $1.7 \times 10^4$ kg C if large algae are included). These biomass assessments increase to $10^2$ to $10^5$ kg C under the maximum observed throughflow velocity.

Using the mean hydraulic conductivity and the allometric biomass conversion, we calculate that over the entire 2014 ablation season 0.3 to 1.1 kg C $km^{-2}$ of microbial cellular carbon was delivered from the supraglacial drainage network to the subglacial environment in western Greenland. However, weathering crust development and biomass transport are defined by the cumulative shortwave radiation receipt and the length of the bare-ice melt season. Therefore, to contextualise our observations in 2014, a typical or average melt year, we recalculate our biomass fluxes for 2006 and 2012: anomalously low and high melt years. Using the mean throughflow velocity and all the biomass estimates, we derive a flux of between $7.2 \times 10^{-4}$ kg C $km^{-2}$ and 0.5 kg C $km^{-2}$ for the 2006 low melt year. For 2012, which sustained high melt, these estimates increase fourfold to $3.1 \times 10^{-3}$ kg C $km^{-2}$ and 2.1 kg C $km^{-2}$. Refining these biomass export estimates on a by-catchment basis, we find a mean of 9.8 kg C $km^{-2}$ and a maximum of 21.8 kg C $km^{-2}$ in 2014, with

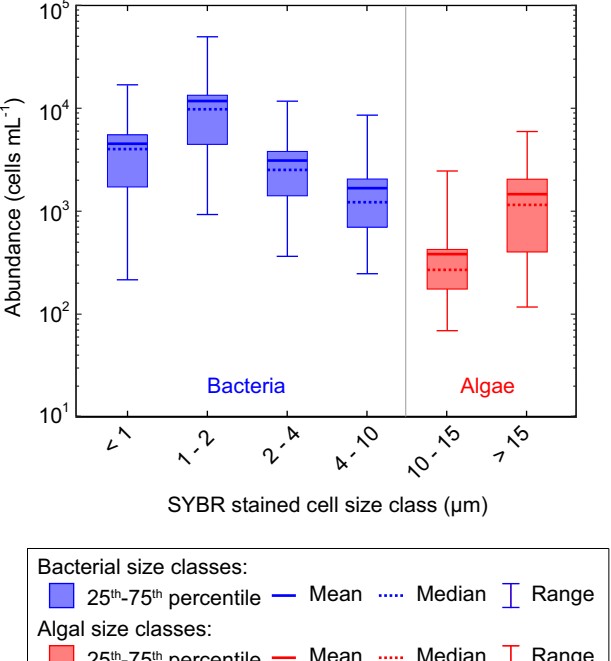

**Fig. 4 Microbial abundance in weathering crust water samples according to cell size.** Microbial size fractions and summary statistics for $n = 73$ independent meltwater samples drawn from the saturated zone within the weathering crust; the nominal classes of bacteria and algae are shown in blue and red, respectively. Samples with total microbial abundance >$10^5$ cells mL$^{-1}$ were excluded ($n = 10$).

4.1 kg C km$^{-2}$ and 16.6 kg C km$^{-2}$ in 2006, and 15.9 kg C km$^{-2}$ and 23.6 kg C km$^{-2}$ in 2012; inclusion of large algae more than doubles these seasonal biomass flux values.

The wide range of values in our assessment of cellular carbon delivery from the weathering crust emphasises its dependence on the melt season duration and intensity, the near-surface hydraulic conductivity of glacier ice, the representative microbial abundance in transport, the biomass conversion utilised, and the treatment of larger algal size fractions. Moreover, the definition of the supraglacial stream network that transports the cells released from the weathering crust also influences these estimates; a simple linear relationship exists between the seasonal mass of cellular carbon entrained and the proportion of a catchment that is defined as stream bank, with a halving of stream-adjacent area leading to a 50% reduction in the carbon biomass transported to downstream environments. We focus our attention on biomass efflux from the saturated weathering crust, which does not account for meltwater and microbe transport through the unsaturated portion of the near-surface ice and, therefore, represents a baseline. Our calculations also assume that the weathering crust is not responding dynamically over synoptic time-scales[48] and, hence we report a cellular carbon liberation to serve as a benchmark that invites future refinement through further investigation of the hydrological configuration, dynamics and functioning of the supraglacial interfluve environment.

**Microbial accumulation in the weathering crust.** Our low observed throughflow velocity of 0.28 m d$^{-1}$ coupled with a mean distance-to-stream index across our catchment of $7.24 \pm 6.6$ m (Fig. 1b), suggests microbes will experience extended transit times through the weathering crust photic zone. As this transit time exceeds typical 4 d doubling times determined for bacteria[50] and algae[58] in glacial meltwaters, we propose that the weathering crust

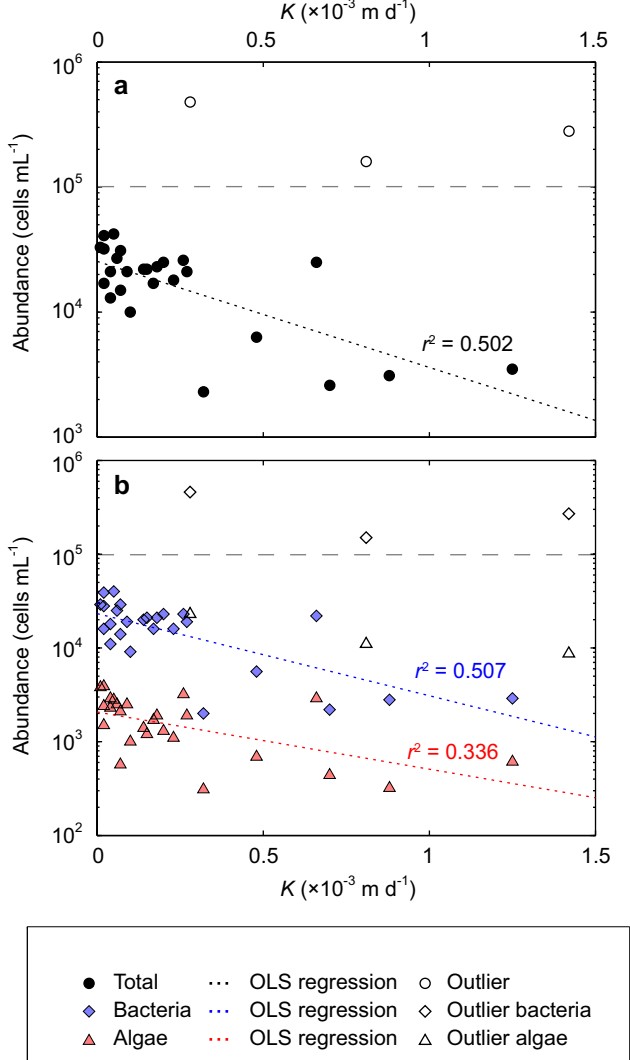

**Fig. 5 Relationships between microbial abundance and hydraulic conductivity ($K$). a** Scatter plot of hydraulic conductivity ($K$) and total microbial abundance for $n = 29$ successful, independent recharge experiments showing ordinary least squares (OLS) exponential regression relationship and coefficient of determination ($r^2$) excluding the outlying samples ($n = 3$) with >$1 \times 10^5$ cells mL$^{-1}$ (see Methods) shown with hollowed markers above a dashed line. **b** Scatter plot, as in **a**, of $K$ against abundance for the size-defined bacterial and algal classes, again indicating the non-linear OLS regression line and $r^2$, and outlying samples indicated with hollow markers.

is a locus for microbially-driven biogeochemical processes, carbon and nutrient transformations, and community growth. The ablating ice surface continually replenishes the weathering crust with emergent and deposited microbes, and there is an ongoing nutrient supply associated with both aeolian-derived and emergent mineral dusts. Solute-rich water films surrounding individual ice crystals are also actively replenished throughout the summer ablation season[53].

A range of microbial biomass doubling times from 1 to 5.5 days[12,34,50,58] have been derived for supraglacial habitats, with 11 days determined for the photic zone community sampled from a Alaskan glacier's weathering crust[35]. Taking the lowest concentration of cells observed in recharge waters ($2.3 \times 10^3$ cells mL$^{-1}$) as a conservative estimate of antecedent near-surface microbial load, we calculate microbial abundance increases of

**Table 1 Catchment-wide microbial abundance and carbon biomass estimated using bacterial and algal cell constants, allometric and constant ratios, and using published supraglacial community doubling times over a single day residence time.**

| | | Cells per unit area $(m^{-2})$ | Carbon equivalent $\times 10^{-3}$ kg C km$^{-2}$ d$^{-1}$ for cells $\leq 15\,\mu m$ (values in parenthesis include larger $>15\,\mu m$ algae) | | | |
| --- | --- | --- | --- | --- | --- | --- |
| | | | Constant cell mass $11+153$ fg[34,50,94] | Constant cell mass $20+260$ fg[83,84] | Allometry[52] | Constant biovolume ratio[86,88] |
| Weathering crust meltwater: | Minimum | $1.46 \times 10^8$ | 2.14 (3.48) | 3.81 (6.1) | 361 (1268) | 432 (1405) |
| | Mean | $14.6 \times 10^8$ | 21.3 (34.6) | 37.9 (60.6) | 3599 (12604) | 4298 (13970) |
| Throughflow export: | $V_t = 0.03$ m d$^{-1}$ | $0.41 \times 10^8$ | 0.08 (0.14) | 0.15 (0.24) | 14.2 (49.6) | 16.9 (55.0) |
| | $V_t = 0.13$ m d$^{-1}$ | $1.86 \times 10^8$ | 0.38 (0.62) | 0.68 (1.08) | 64.4 (226) | 76.9 (250) |
| In situ biomass accumulation, assuming published doubling times (d): | 1 d[34] | $22.9 \times 10^2$ | 2.14 (3.48) | 3.81 (6.09) | 361 (1268) | 432 (1405) |
| | $1+4$ d[34,58] | $21.4 \times 10^2$ | 1.58 (1.84) | 2.85 (3.31) | 71.4 (251) | 119 (312) |
| | 4 d[50,58] | $4.34 \times 10^2$ | 0.40 (0.66) | 0.72 (1.15) | 68.5 (240) | 81.8 (266) |
| | $11+4$ d[35,58] | $1.72 \times 10^2$ | 0.19 (0.44) | 0.34 (0.75) | 51.1 (214) | 57.6 (232) |
| | 11 d[35] | $1.49 \times 10^2$ | 0.14 (0.23) | 0.25 (0.40) | 23.5 (82.4) | 28.1 (91.4) |

$1.5 \times 10^2$ to $2.3 \times 10^3$ cells mL$^{-1}$ d$^{-1}$ using published doubling times. This translates to between $1.4 \times 10^{-4}$ and 1.4 kg km$^{-2}$ d$^{-1}$ of accumulated organic carbon (i.e., the increase in cell numbers) in the weathering crust: the minimum estimate assumes 11 fg per cell, and an 11 d doubling time (excluding large algae), while the maximum includes $>15\,\mu m$ algal cells, and assumes a 1 d doubling time with constant biovolume ratios (see Table 1).

Numerous factors affect supraglacial microbial growth, such as exposure to sunlight, nutrient and water availability[58,59]. In the absence of a well-constrained logistic growth curve for both bacterial and algal communities, we apply a 4 d doubling time to derive a conservative allometric best estimate of cellular carbon accumulation of 0.07 kg km$^{-2}$ d$^{-1}$ (or 0.24 kg km$^{-2}$ d$^{-1}$ if large algae are included) assuming that optimal conditions persist throughout the average melt-season (Table 1). This value is lower, as would be expected, yet comparable with estimates of microbial productivity for the phototrophic algal community at the ice surface (0.23–0.36 kg C km$^{-2}$ d$^{-1}$)[16,58] and mineral-rich cryoconite that typically covers up to 8% of the ice surface (8.7 kg C km$^{-2}$ d$^{-1}$)[16]. It is important to note that these accumulation rates do not include the antecedent concentrations, long-residence time, or emergent populations. Hence, with equivalent allometric biomass efflux projections of only 0.01 kg C km$^{-2}$ d$^{-1}$ (or 0.05 kg C km$^{-2}$ d$^{-1}$ including large algae), our analysis demonstrates the potential for cellular biomass accumulation within the near-surface weathering crust. This hypothesis is further emphasised by coupling our weathering crust data with published algal and cryoconite productivity estimates for the location[16,58] to account for the three constituent supraglacial interfluve environments. We assume that 35% of the bare-ice surface is characterised by active supraglacial streams, as found in our study catchment, with the remaining interfluvial area occupied by the weathering crust and algae, and exhibiting a cryoconite coverage of 8%. This yields near-surface biomass accumulation estimates ranging from 0.59 to 0.82 kg C km$^{-2}$ d$^{-1}$, which far exceed the projected maximum efflux of 0.25 kg C km$^{-2}$ d$^{-1}$ for the site. Consequently, the weathering crust theoretically undergoes a substantive gain in cellular carbon over the melt-season.

**Implications for the carbon cycle of the Greenland Ice Sheet.** Our observations of microbial abundance at the S6 site resonates with those previously reported elsewhere for supraglacial meltwaters[34,35,49,50] confirming that the ice sheet's weathering crust is a biologically-active habitat. Our study catchment suggests that the near-surface weathering crust in western Greenland exhibits an in situ storage of $1.5 \times 10^{15}$ cells km$^{-2}$, equivalent to

between 0.02 and 14.0 kg C km$^{-2}$, depending on the biomass conversion applied. This near-surface habitat is hydrologically active and characterised by protracted residence times owing to slow transport of meltwater through the spatially extensive porous interfluve area. To improve models of near-surface microbial transport, it is essential to explore the microbe entrainment and transport-controlling environmental characteristics (such as local ionic strength, water pH, pore space configuration and roughness, as seen in other porous media[60]) and biophysical responses[54–56]. For example, the crystal size and dust content of Greenland's ablation area is known to vary, in part due to the era when the emergent ice was formed[61]; such properties will influence spatial patterns in surface hydraulic conductivity and microbe entrainment. Nonetheless, the mean near-surface weathering crust hydraulic conductivity measured across the study catchment, during a typical Greenland melt season[62], compares well to other supraglacial evaluations[36], and yields a specific daily microbe flux of only $4.1 \times 10^7$ cells. Here, we expand on three core implications of this low rate of near-surface cellular biomass transport: the fluvial export of biomass from the ice sheet surface, the accumulation of biomass in the weathering crust, and the associated supraglacial carbon cycling.

Across an area of $\sim 14 \times 10^3$ km$^2$ in western Greenland during 2014, our allometric best-estimate of the ablation season biomass export from the surface weathering crust to moulins descending into the ice sheet interior is 3.8 to 15.5 tonnes of cellular carbon. This highlights the importance of the flushing of organic matter from the ice sheet's surface to the subglacial environment. In low-melt years, these biomass estimates can decrease by an order of magnitude, while in high-melt years the values can double. However, current understanding of the nature of meltwater routing, transit times and the redox conditions in Greenland's subglacial environment is incomplete. Therefore, we propose that the transport and potential subglacial deposition and storage of cellular organic carbon and associated compounds by inefficient basal drainage networks enhances microbe-water-rock interaction times and ratios, and provides a viable and contributory source of carbon for the support of heterotrophs[23] and for methanogenesis[63] at the ice sheet bed. Moreover, with knowledge that at least a portion of the weathering crust communities are active[51,56,58], through their export from the supraglacial environment, they may deliver functionality to, and inoculate downstream sub- and pro-glacial aquatic systems. It is also established that the weathering crust evolves and decays, respectively, under clear-sky and cloudy or rainfall-dominated conditions[48]. Consequently, the synoptic and seasonal patterns, and future of

supraglacial cellular biomass export warrant further investigation given the increasing clear-sky ablation season phases[64], expanding bare-ice area[20] and/or more frequent rainfall events[65] forecast for Greenland under future atmospheric warming.

The abundance and proliferation of microbial cells across the bare-ice ablation zone is dependent on the duration of the melt season and cumulative shortwave irradiance[14]. Our findings highlight how cell export from the ice surface is also conditioned by these environmental variables indicating that, during high melt years, two competing amplification mechanisms interact at seasonal time-scales: in situ microbial biomass accumulation and hydrological export both increase. Nonetheless, our assertion that cellular biomass export lies below the equivalent cellular load or estimated accumulation is of particular significance given the biological-darkening of the ice sheet (bio-albedo), where microbial abundance strongly influences the bare-ice reflectivity, darkens surface ice and enhances surface melt cycles[9,11–14,58]. Studies suggest that the darkening phenomenon on the ice sheet is driven by predominantly phototrophic algal growth in the top few centimetres of the ice sheet[12–14,58]. Our results indicate the potential for autochthonous microbial carbon accumulation in this shallow near-surface weathering crust; such accumulation also contributes to this biological darkening of the ice. Variability in the accumulation and export of cellular biomass, respectively, within and from the weathering crust can contribute to the strong inter- and intra-annual variation in observed ice sheet albedo[38,66].

The disparity between the accumulation and export of cellular biomass across the western sector of the ice sheet also imparts a number of biogeochemical cycling processes. For example, with cryoconite holes punctuating[48] and hydraulically linked[44] to the porous weathering crust, it is unknown whether these features represent biomass sinks where a proportion of the entrained microbes within the near-surface become bound to cryoconite granules[55] and are removed from the water column. Carbon cycling within the weathering crust itself is also promoted through cell mortality, from grazing by protists, viral lysis and photolysis[34]. Indeed, the cycling of organic carbon and exudation or release of dissolved organic carbon (DOC) by microbes is commonly reported in supraglacial meltwaters[3,15,23,67,68] and is evidenced by elevated DOC in near-surface ice[69]. There will likely be a preference for the cycling of new, labile, rather than older, recalcitrant, 'fossil' carbon[70]. Moreover, the extended residence time of microbes within the weathering crust can also account for DOC consumption[56]. Combined, these processes will modulate the organic carbon and associated compounds supplied to subglacial and downstream environments. Given geophysical evidence of saturated sediments at least 1 m thick beneath the western Greenland Ice Sheet[71,72], coupled with indications of subglacial methane cycling[63] and the emergence of methane-saturated proglacial waters[4], our findings confirm a considerable organic carbon flux enters and is likely sequestered and/or transformed within the subglacial environment. Consequently, there remains a need to better constrain microbial carbon cycling pathways and their controls across supraglacial, subglacial and proglacial environments in Greenland.

## Methods

**Study location.** Our study site was located proximate to the S6 automatic weather station (AWS) on the Kangerlussuaq Transect in western Greenland (67° 04.78'N, 49° 24.08'W), 38 km from the ice margin at an elevation of approximately 1020 m a.s.l. within the well-reported Dark Zone (Fig. 1a). The location has a mean annual air temperature (2003-2016) of $-10.2\,^{\circ}C$, which has shown a rising trend over the last decade[73]. Observations over the last two decades suggest an annual ablation at the site of $-1.96$ m w.e.[62]. Situated on the western edge of the Dark Zone, a combination of enhanced dust content, black carbon and microbial community loading reduce the local bare-ice albedo at the site[12,74,75]. Neighbouring ice temperature observations[76] show that at ~1000 m a.s.l. only the uppermost 1–2 m of

the western sector of the ice sheet's ablation zone reaches $0\,^{\circ}C$ during the summer ablation season. This shallow, seasonally transient, temperate ice layer is likely to develop as a porous and hydrologically active weathering crust during summer[31].

**Meteorology and meltwater production.** Hourly meteorological data from the S6 AWS[73] were used as input to a point-based energy balance model[77] to estimate local melt conditions across the low elevation range supraglacial catchment during the observation period. We estimated ice surface albedo for the catchment simplistically using a 12-hour running mean of the ratio of incoming and outgoing shortwave radiation at the S6 AWS, filtered for erroneous values, and which ranged between 0.37 and 0.50. The surface aerodynamic roughness parameter ($z_0$) was kept constant, and taken as an average derived from a set of ten random 10 m transects[78] oriented perpendicular to the dominant south-easterly wind (mean orientation: 116°) during the field campaign with elevation data extracted from the 0.25 m resolution DEM (see below, and Supplementary Method S1) within the supraglacial catchment of interest: $z_0 = 1.142$ mm compared well to the $10^{-3}–10^{-4}$ m range quoted for bare-ice in the locality[79]. The 2014 summer season was unexceptional in surface mass balance terms, suggesting our study period provides a representative baseline[62,73].

**Near-surface hydrological functioning.** Observations of the weathering crust and its hydrological functioning were made following Stevens et al.[36]: briefly, shallow (0.26 or 0.36 m) holes were made using a 50 mm diameter Kovacs ice auger, and bespoke capacitance piezometers were used to record bail-recharge experiments, yielding water level recovery records at 2 s intervals. Hydraulic conductivity ($K$) was assessed from these recharge curves following standard groundwater techniques[36] (see Supplementary Method S2). A total of 9 experimental auger hole sites, separated by distances of ~9 m, were located in a quasi-random grid aligned perpendicular to the primary channel flowing to the northwest (Fig. 1a). Auger holes were evacuated manually using a BiOrb™ syphon after the syphon was rinsed and flushed three times with supraglacial stream water; similarly, the piezometers were rinsed three times prior to installation and onset of recharge. Variance in recharge rates made systematic timing of experiments across the experimental grid unfeasible. A portion (c. 40%) of all the recharge experiments undertaken bore incomplete recharge, which precluded confident estimation of $K$-values[36].

**Microbial enumeration.** Following auger hole recharge experiments, using a polyethylene syringe and 30 cm polypropylene tube rinsed three times in supraglacial stream water prior to sampling, a 15 mL depth-integrated sample of recharge water was abstracted. Of this, 10 mL was decanted into a 15 mL sterile polyethylene centrifuge tube, and fixed using 50 μL glutaraldehyde (2% w/v final concentration). The preserved samples were kept dark and cool (~4 °C) for up to 8 days while in the field and in transit, and subsequently fast frozen and stored at $-80\,^{\circ}C$ until analysis.

We used flow cytometry[10,49,80] to enumerate the microbes in the recharge water samples, employing the following protocol: samples were thawed at ambient laboratory room temperature, gently agitated, and stained with SYBR Gold (Molecular Probes, UK) at a final concentration of 1× and stored in the dark at 20 °C for a maximum of 240 minutes prior to analysis. Enumeration was conducted with a Sony SH-800EC Cell-Sorter (Sony Biotechnology, Japan), using gates set within the Sony Cell Sorter v.2.1.3 software package to describe stained and non-stained particles as optimised for the sample water type[15] (see Supplementary Fig. 1). Low concentrations of suspended mineral particles returned an estimated uncertainty in microbial abundance estimates of ~10% ref. [81]. Cell size categories ranging from <1 to >15 μm were defined using a non-fluorescent Flow Cytometry Size Calibration Kit (Molecular Probes, UK) following the manufacturer's instructions (see Supplementary Method S3). Our size assessment was constrained by the instrumental ~0.5 μm analytical limit, with a threshold applied to eliminate detection noise.

To remain conservative in our assessment of the weathering crust hydrology, recharge water samples yielding microbial abundance of >$1.0 \times 10^5$ cells mL$^{-1}$ were treated as outliers. These ten samples, we found, were proximate to highly localised microbial blooms and/or anomalously high particle concentrations, as reported by other authors[10,12,51,58]. We justify our approach here based on cell abundances of <$10^5$ mL$^{-1}$ being distinctive of supraglacial meltwaters[49,50], weathering crust water[35], and glacier ice[82]. Further, eighteen ancillary water samples taken from exploratory auger and cryoconite holes in bare-ice areas at the S6 site demonstrated no cell concentrations >$10^5$ mL$^{-1}$.

We employed a 10 μm size threshold to nominally classify microbes as bacteria or algae[37]. The microbes classed as algae, typically accounted for <10 % of the observed abundance. Given the filamentous nature of many cyanobacteria in glacial environments[55], it is likely that a proportion of the cyanobacteria were included in our algal category. Because viruses, filamentous cyanobacteria and ice algae were not specifically identified, differentiated, or excluded by gating in our approach, uncertainties lie in the classification (and therefore biomass) of microbes particularly in the lower- and upper-most size fractions reported here.

**Biomass estimation**. Microbial abundance can be converted to carbon biomass using three established techniques[52]: (i) constant carbon content, (ii) allometric, or (iii) constant carbon ratio approximations. Previous work has suggested constant values for bacteria and archaea (e.g., 11 fg C cell$^{-1}$ (ref. [20,34]) fg C cell$^{-1}$ (ref. [83])) or equivalent estimates for cyanobacteria and algal cells (e.g., 153 fg C cell$^{-1}$ (ref. [84])). However, these approaches may not fully reflect variations in microbe size. Consequently, biomass can also be approximated using either a size-dependent allometric model[85] or a constant biovolume ratio[86].

Here, owing to the wide variety of unknown microbe geometries, biovolumes (V) were estimated by assuming bacteria <1 μm are spherical, with larger bacteria as rods (hemispherical-ended cylinders) exhibiting length:width ratios defined by typical bacterial geometries[37,85]; algae were described as simple 10 μm diameter cylinders[14,58,87]. For each of these three geometries, the cell length was defined by the midpoint of the size fraction reported by the cytometric analysis. Allometric methods ascribe the mass of carbon (M, in fg C) as a function of V (in μm$^{-3}$) and scaling factors (c and a):

$$M = cV^a \tag{2}$$

in which, for bacteria, $c = 162$, $a = 0.91$ (ref. [85]); and for algae, $c = 109$, and $a = 0.991$ (ref. [84]). Norland[52] argued that the allometric biomass method provides the most robust calculation of biomass, and is therefore referred to as our best estimate. Derived constant biovolume ratios assessed for bacteria and algae indicate, respectively, 560 fg C μm$^{-3}$ (ref. [86]) and 110 fg C μm$^{-3}$ for algal plasma volume[88].

Despite the potential for including large viruses in the <1 μm cell counts, the size fraction was included in the biomass estimates based on the existing evidence of such small bacteria within Greenland's ice[89]. However, due to the uncertainty over the size of algal cells or filaments in the >15 μm category owing to the limitations of the FCM analysis, these were ascribed a nominal mean length of 27.5 μm, which equates to that typical of an *Ancylonema nordenskiöldii* alga, common to the locality[14]. These algal contributions are included as accompanying data in our carbon calculations given their capacity to exaggerate the biomass estimates.

To contextualise our biomass results, and in recognition of the extent of the Dark Zone and presence of elevated abundances associated with emergent dusts and algal blooms across the region, we re-ran our calculations using the mean cell concentration and size distributions derived from our entire sample set. The inclusion of outlying samples increased the mean abundance by less than a factor of 3, while the difference in proportion of cells in each size fraction was ≤1%.

**Supraglacial catchment characterisation**. To characterise the supraglacial catchment, overlapping digital images acquired by a fixed-wing UAV were used to yield high resolution orthoimagery and DEM of the study site. Full details of the UAV and protocol followed to derive the spatial data sets are given in our Supplementary Method S4 and Ryan et al.[11]. Briefly, 740 overlapping RAW format images were acquired at 350 m above the ice surface using a Sony NEX-5N camera. These images were georeferenced with on-board GPS and UAV altitude data and processed into orthomosaics and DEMs using Agisoft's PhotoScan Pro. The resulting orthomosaic and DEM products yielded a horizontal ground resolution of ~0.1 m (Fig. 1a).

Due to the absence of constrained ground control points on the ice surface to geolocate the processed DEM, and uncertainty associated with the onboard GPS and limited area-of-observation relating to the flightpath, the DEM contained well-known photogrammetric artefacts in its 3D geometry[90] resulting in hydrological inconsistencies. To address these artefacts, the photogrammetric DEM was co-registered to the 2 m horizontal resolution ArcticDEM (https://www.pgc.umn.edu/data/arcticdem/)[91] using 75 control points manually located in both images. A thin-planted spline warping was applied to the UAV-derived DEM. To correct the hydrological inconsistencies the two DEMs were resampled to a resolution of 0.25 m and detrended using 31 × 31 plane fitting filter in the RSGISLib software[92]. The detrending process yielded two raster surfaces, the first high frequency signal (e.g., local stream channels and large cryoconite holes) and the second, the low frequency signal (i.e., the underlying elevation surface). To generate the final DEM the high frequency signal from the UAV was combined with the low frequency signal from the ArcticDEM, providing a hydrologically consistent DEM, and an 11 × 11 kernel, two-standard-deviation Gaussian smoothing filter was applied. The combined, tuned elevation model was then resampled to a 1 m horizontal resolution using a cubic spline.

To estimate supraglacial drainage, a surface flow routing method that accounts for features causing hydrological fragmentation is a viable protocol[93]. Analyses of the DEM and ortho-imagery within ESRI's ArcGIS v.10.5 Spatial Analyst indicated that abundant, large cryoconite holes were associated with local topographic depressions or sinks. Therefore, sinks within the DEM were filled using a threshold depth of 0.51 m based on the maximum depth recorded in a survey of 90 random cryoconite holes with diameters ranging from 0.01 to 1 m at the study site. Sinks >0.51 m in depth were retained, to account for any crevasse or fracture features, despite their relative paucity in the local catchment area.

Standard flow direction (d8) algorithms were used on the filled DEM to derive flow accumulation and stream network maps. Cross-examination of these with the high-resolution UAV ortho-imagery indicated a high degree of fit, and facilitated identification of a flow accumulation threshold of 165 m$^2$ to define active supraglacial channels. This threshold returned a ~0.5 km$^2$ supraglacial catchment

containing our experimental plot area, with elevations increasing up-glacier by ~20 m over a 2.5 km distance and a mean pixel surface slope of 2.23 (±1.70)%. Following the GIS-based identification of the primary stream network based on the flow accumulation threshold, the streams were removed or 'burned'[28] from the DEM and the hydrological assessment to calculate supraglacial interfluve flowpath lengths (Fig. 1b). Sensitivity testing using contrasting parameters in our workflow demonstrated that predicted hydrological variables were robust. For example, a 10% change in the accumulation area used to define the stream network resulted in only a 5% change in the average distance-to-stream metric, and in the resulting throughflow velocities. However, as noted by Yang et al.[28], the uncertainties associated with DEM registration and resolution, the threshold stream size used to define the supraglacial network and, here, the geometry of the saturated zone and subsurface flow that characterises the weathering crust, require further interrogation which is beyond the scope of this paper.

**Catchment upscaling**. Given the absence of a clear relationship between melt rate and $K$[36], to reveal a regional picture we upscaled our observed data across 795 moulin-terminating supraglacial catchments in western Greenland defined by Yang and Smith[22], employing the bare-ice duration maps from Ryan et al.[20] for 2006, 2012, and the study year 2014. The mean summer season (June to August: JJA) bare-ice duration ($b_i$) for each catchment was used to account for latitudinal and elevational gradients, and annual variability. We presume that the depth of the hydraulically active weathering crust evolved over time: deepening from the onset of bare-ice, reaching a maximum depth (≤0.29 m), and declining thereafter. We used a modified cosine function centred on July 15 (day of year (DOY) 196) and defined by our observation period to describe this growth and decay of the weathering crust: for short bare-ice durations ($b_i$ <38 days), the hydraulically active layer did not reach the 0.29 m maximum but reaches its greatest depth of 0.0076·$b_i$, while for longer bare-ice durations, the depth increases to 0.29 m and plateaus at that maximum prior to decaying (see Supplementary Method S5). From this simple evolving depth model and our mean microbial abundance, for each catchment within the $1.38 \times 10^3$ km$^2$ region, we calculated a cell flux to supraglacial streams assuming a constant proportion (14%) for stream bank area over each of the internally drainage catchments, as found in our study catchment. We assume that rapid (<1 d) in-stream transit time[22,45,46] from source to moulins results in minimal change in microbial abundance and biomass during transport.

**Reporting summary**. Further information on research design is available in the Nature Research Reporting Summary linked to this article.

## Data availability

The datasets generated during and/or analysed in this study are available in the Zenodo repository (https://doi.org/10.5281/zenodo.4623697). Source imagery files from the UAV collected on 8 August 2014 are archived in the Pangaea repository (https://doi.org/10.1594/PANGAEA.885798) and available from the corresponding author on reasonable request. The energy balance model is available on Zenodo (https://doi.org/10.5281/zenodo.3228331). ArcticDEM data is available via https://www.pgc.umn.edu/data/arcticdem/ and RSGISLib at https://www.rsgislib.org/; the S6 weather station data is available on request from the Institute for Marine and Atmospheric Research of Utrecht University.

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

## Acknowledgements

This research was supported by a Royal Society Grant (RG130314: PRESTIGE, to A.E. and T.D.L.I.-F.). We acknowledge NERC Large Grant (NE/M020991/1 and NE/M021025: Black & Bloom, supporting T.D.L.I.-F., J.M.C. and C.J.W., with recognition of project investigators including M Tranter, AJ Hodson, E Hanna, and M Yallop). T.D.L.I.-F. acknowledges Leverhulme Trust Fellowship RF-2018-584/4. A.E. acknowledges Leverhulme Trust Fellowship RF-2017-652. A.E., T.D.L.I.-F., S.M.E.R. and J.M.C. all recognise NERC Standard Grant (NE/S001034/1: MicroMelt). A.C.M. and A.E. acknowledge support from a National Research Network for Low Carbon Energy and Environment (NRN-LCEE) grant from the Welsh Government and the Higher Education Funding Council for Wales (HEFCW): Geo-Carb-Cymru. J.M.C. recognises the Rolex Awards for Enterprise and National Geographic. K.A.C. acknowledges funding from the European Union's Horizon 2020 research and innovation programme under the Marie Skłodowska-Curie Grant agreement No. 663830. Financial support was also provided to K.A.C. by the Welsh Government and Higher Education Funding Council for Wales through the Sêr Cymru National Research Network for Low Carbon, Energy and Environment. K.N. was involved through a SNSF Mobility Fellowship Grant (P2FRP2/174888); A.H. acknowledges a research professorship from the Research Council of Norway through its Centres of Excellence scheme (Grant 223259) and an Academy of Finland ArcI visiting fellowship to the University of Oulu. Logistical field support from the Dark Snow Project (www.darksnow.org) led by J.E.B., and a Villum Young Investigator Programme (Grant VKR-023121) and a Czech Science Foundation (Grant: 19-21341 S) held by M.S. are gratefully recognised. Meteorological data from the S6 AWS maintained by Utrecht University, Institute for Marine and Atmospheric Research Utrecht (UU/IMAU) were kindly made available by M.R. van den Broeke and C.J.P.P. Smeets. We thank J.W. Bridge for constructive input.

## Author contributions

T.D.L.I.-F. conceived and designed the study, conducted the experimental fieldwork and analyses, and wrote the manuscript; A.E. contributed as lead of 2014 fieldwork phase, and to analytical development; I.T.S. established the flow cytometry protocol and derived data for K and microbial abundance in collaboration with T.D.L.I.-F., A.E. and A.C.M; A.H. and J.R. led the collection and pre-processing of the UAV datasets and contributed to the upscaling analysis; K.N. and P.B. contributed to post-processing of the imagery and refinement of spatial datasets; J.M.C. contributed fieldwork assistance and input to biomass comparisons; C.J.W. and S.M.E.R. contributed to biomass calculations and comparisons; K.A.C. contributed logistical support for the field campaign led by J.E.B. and M.S; A.H., A.E., I.T.S. and A.C.M., and all other authors contributed to substantive editing and finalisation of the paper.

## Competing interests

The authors declare no competing interests.
