## [Peer Review File · Nature Communications]

REVIEWER COMMENTS

Reviewer #1 (Remarks to the Author):

The manuscript reports the export of microbial biomass from the Greenland Ice Sheet. From the cell flux, the authors calculate the resulting flux of carbon from the ice sheet. The authors also examine biomass accumulation in the weathering crust where higher biomass accumulate and carbon cycling is enhanced. The authors conclude by quantifying the total carbon flushed from the GrIS and posit that this amount of carbon would support heterotrophy and methanogenesis in the subglacial environment.

Introduction:

I found the introduction to be well-written - concise and properly cited with an appropriate amount of detail. I'm not overly familiar with weathering crust and appreciated the description in the introduction.

Results:

I appreciate the inclusion of different cell sizes in the analyses - seems particularly informative for these systems that are mixed microbial and microeukaryotes.

Page 4: 'bacteria' and 'archaea'?

Discussion:

The discussion is also quite nice. The data are summarized and placed into context with previous studies. The authors are careful to place upper and lower constraints on their data so as to not overstate or over-speculate given the heterogeneous nature and the difficulty in scaling across the larger ice sheet.

Materials and Methods;

The methods are well described and cited. I am curious how representative a 10-mL sample is to then scale up. I suppose this small sample size is made more robust by including a number of auger sites.

Page 15: 'bacteria' and 'archaea'?

Figures:

Figure 2: Can you remind the reader in the legend why there is a dashed line at 10^5 on the panel e. (same comment for figure 3)

Figure 4: As noted above, this archaea should also be included.

Reviewer #2 (Remarks to the Author):

The paper studies microbial carbon production in the near-surface weathering crust photic zone at the Greenland Ice Sheet (GrIS) and its potential discharge to the subglacial domain via supraglacial meltwater. Estimated discharge rates and carbon concentrations over an assumed annual melting period of 70 days are scaled from the study area (0.36 km²) to an area covering the Western part of the GrIS (2.3 x 10⁴ km²). Presented data suggest an uncertainty range in seasonal carbon export to

the ice sheet bed in the range between 130 kg to 400 ton, perhaps up to a declared speculative 950 tons considering the potential for algal blooms and dust.

The presented results are novel and should be of interest to both the primary target group (i.e. research community studying microbial processes), as well as to the broader scientific community studying downstream processes. The presented results and conclusions are well presented, referenced and documented, with adequate descriptions of used methodologies, assumptions and uncertainty ranges.

My assessment of the paper is that it is a novel, well-written and scientifically sound piece of work that is suitable for publication within the scope of Nature Communications.

However, before publication, I recommend that the manuscript is revised taking the following two minor issues into consideration:

1) The title of the paper does not imply that the study covers only a part of the Greenland Ice Sheet, which spans several climatic regions, with likely differences in the meltwater regime. Also, the title does not include the work on estimating the in-situ production of microbial carbon in the weathering crust which forms an essential part of the study. I recommend that the title is updated to more precisely reflect the area of the study domain (i.e. Western part of the GrIS) and that production is also included. A suggestion for a revised title could be "Production, storage and export of cellular carbon from the Western part of the Greenland Ice Sheet"

2) The abstract states that the authors estimate that up to 10^3 tonnes of cellular carbon is exported from the surface. This estimate is equal to the upper limit of the speculative guess (as mentioned above) and does not reflect the rather large uncertainty range that the data from the study suggest (i.e. from 130 kg to 950 ton carbon). I recommend that the wording of the abstract is rephrased to include this important uncertainty range, which also provides a clear argument for the need for further studies to constrain this uncertainty.

Reviewer #3 (Remarks to the Author):

Please note that my expertise is in ice sheet hydrology, rather than flow cytometry enumeration and biogeochemistry.

This is an interesting and well-written paper that presents several noteworthy results. Firstly, that accumulation of cellular carbon in the near-surface photic weathering crust exceeds fluvial export for the study's supraglacial catchment of the Greenland Ice Sheet. This is based on typical inter-stream distances, the speed of water flow in the weathering crust, and typical doubling times from other supraglacial environments. This near-surface accumulation of carbon would have important implications for the biological impact on the ice surface albedo (so-called 'bio-albedo'). This finding would, however, be much more robust if it were based on measurements of carbon in the weathering crust over the full 70-day melt season, rather than hydraulic conductivity and previously published cell accumulation rates.

Secondly, the authors extrapolate their carbon flux data to the ablation area of the western Greenland Ice Sheet margin over an estimated 70-day bare-ice melt season. These up-scaled results indicate that between 400 and 950 tonnes of carbon is delivered to the ice sheet bed, and the majority of this reaches downstream ecosystems. Note that later (at the end of page 10) the authors mention that "seasonal biomass export to moulin-terminating supraglacial streams is 22 - 77 tonnes of cellular carbon", but it is not immediately clear to what temporal and spatial scales these numbers refer. The field methods employed seem robust and the resulting data of high quality (in terms of ice sheet hydrology and DEM generation). My main (and I think a critical) issue is the extrapolation of data from a single small supraglacial catchment over 6 days, to a large part of the western margin of the

Greenland Ice Sheet over a 70-day melt season. The upscaling method is not fully or clearly explained (there is just a reference to Yang and Smith 2016). In their extrapolation, how do the authors account for temporal and spatial variations in (amongst other things): ice albedo, ice surface melt rate, water table height, inter-stream distance, distribution of moulins and crevasses (which affect the proportion of a catchment that consists of water flow within the near-surface weathering crust compared to in supraglacial streams and rivers)? For such a crucial component of the main findings of the paper (and the result that would appeal to a broad audience), there needs to be a more robust and thorough presentation of the approach utilised.

Specific points (by page number, P – note that it really would be useful if Nature Communications submissions came with line numbers).

P1. (abstract) the authors use a combination of their measurements of hydraulic conductivity in the near-surface weathering crust along with previously published estimates of microbial productivity. I would argue that they therefore do not really 'demonstrate that cellular carbon accumulation in the weathering crust exceeds fluvial export', but rather they infer this.

P1. (abstract) The last sentence of the abstract is really very speculative and relates back to my main criticism of the unexplained extrapolation of results from a single catchment over 6 days to a large part of the ablation area of the western margin of the ice sheet over an estimated 70-day melt season.

P6. "...is hydraulically-active at the ablating margin of the GrIS" is too broad based on the data presented. "...is hydraulically-active at our study catchment 38 km from the ablating margin of the GrIS" would be more accurate.

P6. Might "using" be better than "under"?

P6. Do the authors know what proportion of the inter-stream area in the rest of the ablation area is dominated by the slow water flow through the weathering crust? It is entirely possible that at lower elevations particularly, most of the inter-stream water flow is through rills and micro-channels.

P6. It seems likely that at times in the study catchment, and perhaps more frequently lower in the ablation area, the water table would rise up so that flow was principally through the higher porosity unsaturated ice layer. This would mean that the authors' estimates of nutrient and carbon storage are also time- and space-dependent and cannot be simplistically scaled up to the entire western margin of the ice sheet over a whole melt season.

P8. The high bound of the authors' estimate of carbon export (950 tonnes) seems over speculative – no quantification of the "heightened mean abundance reflective of the dust and algal blooms commonly found across the ice-sheet's south-western margin" is put forward.

P10. The "Implications..." section rests on the unjustified assumption that the study catchment is "broadly representative of the region's supraglacial environment". This is quite an assumption and is unlikely to be justified given known spatial variations in (amongst other things): ice albedo, ice surface melt rate, water table height, inter-stream distance, distribution of moulins and crevasses. No evidence is presented that lends support to this assertion.

P11. "phenomena" should be "phenomenon" I think.

P11. "in the top few centimetres of the ice sheet surface" should be "in the top few centimetres of the ice sheet" I think.

P18. "...define THE supraglacial network..."

P18. There is an order of magnitude range in the estimates of stream spacing, which is key to quantifying the relative importance of slow water flow in the near-surface weathering layer (and thus the degree to which cells and carbon accumulate in this layer). It would be good to see a sensitivity analysis of the carbon storage and export estimations when varying the stream spacing during the spatial and temporal extrapolation.

P20. (Fig. 1 caption) "perceptive" should be "perspective" I think. While the perspective approach does allow the authors to show all three mapped layers in a small space, a more traditional planar display of the data would be clearer and show them in their entirety.

Response to the Editor

Editor comments in black standard text, our responses indicated in italic red text.

As you will see from the reports copied below, while all three reviewers agree that the work is interesting and potentially noteworthy, they raise important concerns about the robustness of some of the extrapolations. We find that these concerns currently limit the strength of the study, and therefore we ask you to address them with additional work. Without substantial revisions, we will be unlikely to send the paper back to review.

We thank the three Reviewers for constructive commentary on our paper. We note that both Reviewer 1 and 2 were very positive with respect to our manuscript and raise more minor suggestions for revisions; we have addressed this in full (see details below). We acknowledge Reviewer 3 raised a potential lack of clarity in our upscaling and over what region in Greenland this was applied. We also recognise our initial approach had been overly simplified, given the upscaling was used simply to present a potential context to our results. In response to this concern, we have completely revised our upscaling approach, and now provide additional Supplementary Information to aid clarity in our approach. In our revised text, we have carefully described the data and catchment areas and employed an entirely new upscaling approach which reflects latitudinal and elevational gradients over western Greenland. This is reported in an entirely reworked section “Microbial export from the weathering crust in western Greenland”. Our approach, while remaining a first order approximation, now includes an evolving weathering crust that is described by the duration of bare ice during the summer months, and presents the biomass liberation in terms of cellular carbon per km². This new methodology reduces our estimates of carbon fluxes to the subglacial environment, but our core message and interpretations remain the same.

A number of the points raised by Reviewer 3 highlight uncertainties in the evolution and functioning of near-surface ice, uncertainties that we described and acknowledged in the original manuscript, but which extend beyond the scope of our paper and demand the future work that our paper seeks to instigate through its contribution. We did, however, find Reviewer 3 appeared to have made an assumption regarding a section of our discussion, and overlooked the indication that its content was specific to our study area and not dependent on the upscaling. We can only reiterate we have been transparent with declaring a wide range of uncertainties in our study – and this approach was commended by Reviewer 1 in terms of not being overly speculative with our quoted values of biomass. Our revised text, throughout our reworked discussions, has endeavoured to clarify these potential misconceptions.

We trust substantial revisions can be seen in: (1) our thorough edits to the sections “Microbial and carbon fluxes”; (2) the entirely revised “Microbial export from the weathering crust in western Greenland” and “Catchment upscaling” sections, which are supported by a newly appended Supplementary Information text; and (3) more minor edits throughout the manuscript which are aimed to improve clarity and relevance in response to points raised by the three Reviewers.

Response to Reviewer comments

Reviewer comments in black standard text, our responses indicated in italic red text.

Reviewer #1

The manuscript reports the export of microbial biomass from the Greenland Ice Sheet. From the cell flux, the authors calculate the resulting flux of carbon from the ice sheet. The authors also examine biomass accumulation in the weathering crust where higher biomass accumulate and carbon cycling is enhanced. The authors conclude by quantifying the total carbon flushed from the GrIS and posit that this amount of carbon would support heterotrophy and methanogenesis in the subglacial environment.

Introduction:

I found the introduction to be well-written - concise and properly cited with and appropriate amount of detail. I'm not overly familiar with weathering crust and appreciated the description in the introduction.

We are pleased to learn the reviewer found our introductory text helpful and adequately detailed.

Results: I appreciate the inclusion of different cell sizes in the analyses - seems particularly informative for these systems that are mixed microbial and microeukaryotes.

We are grateful to the reviewer for highlighting the need to accommodate complexity in the supraglacial ecosystem, and their recognition of our attempt to do so in a justified and appropriate manner.

Page 4: 'bacteria' and 'archaea'?

We have included 'archaea' as appropriate in the text. Note, here, we refer to 'bacteria sized' microbial populations and larger microbes, assumed to be algae.

Discussion: The discussion is also quite nice. The data are summarized and placed into context with previous studies. The authors are careful to place upper and lower constraints on their data so as to not overstate or over-speculate given the heterogeneous nature and the difficulty in scaling across the larger ice sheet.

We are appreciative of the Reviewer's positive perspective on our clear and open approach, and our use of ranges of values given the novelty of this study in a hitherto poorly characterised environment.

Materials and Methods: The methods are well described and cited. I am curious how representative a 10-mL sample is to then scale up. I suppose this is small sample size is made more robust by including a number of auger sites.

We are pleased to see the Reviewer finds our description of methods is clear and appropriate. The observation regarding the 'upscaling' is correct in that we use samples drawn from 9 auger hole sites and over a number of days as a best estimate for the locality. Because of the variable nature of the ice surface and its hydrology, as we noted, a proportion of our recharge experiments failed to provide hydrological conductivity values. Additional data relating to the hydraulic conductivity is reported in Stevens et al (2018: Hydrological Processes) and is already cited here. There seems not to be a request for a revision here from the Reviewer, so no action has been taken.

Page 15: 'bacteria' and 'archaea'?

We have included 'archaea' as appropriate in the text, as suggested. See response above.

Figures:

Figure 2: Can you remind the reader in the legend why there is a dashed line at 10^5 on the panel e. (same comment for figure 3)

We have addressed the Reviewer's comment by inserting additional text in the figure captions (both Figure 2 and 3) to clarify the threshold shown by the dashed line which is explained in the original (unchanged) enumeration Methods section text:

"To remain conservative in our assessment of the weathering crust hydrology, recharge water samples yielding microbe abundance of $> 1.0 \times 10^5$ cells mL^{-1} were treated as outliers. These ten samples, we found, were proximate to highly localised microbial blooms and/or anomalously high particle concentrations".

Figure 4: As noted above, this archaea should also be included.

We are unclear on the correction indicated or requested by the Reviewer here. In our original description of "Results" we had indicated that the 'bacteria' and 'algae' categories were a simplified size-based classification. However, to address this we have reworded this section to improve the clarity:

"To differentiate between bacteria and archaea, and larger algae, a $10 \mu\text{m}$ size classification threshold was applied. Examination of the association between K and abundance independently for these 'bacteria' and 'algae' classes highlights..."

We have further emphasised this 'simplified classification' by noting "the bacterial... and algal... size fractions" in the figure caption.

Reviewer #2:

The paper studies microbial carbon production in the near-surface weathering crust photic zone at the Greenland Ice Sheet (GrIS) and its potential discharge to the subglacial domain via supraglacial meltwater. Estimated discharge rates and carbon concentrations over an assumed annual melting period of 70 days are scaled from the study area (0.36 km²) to an area covering the Western part of the GrIS (2.3 x 10⁴ km²). Presented data suggest an uncertainty range in seasonal carbon export to the ice sheet bed in the range between 130 kg to 400 ton, perhaps up to a declared speculative 950 tons considering the potential for algal blooms and dust.

The presented results are novel and should be of interest to both the primary target group (i.e. research community studying microbial processes), as well as to the broader scientific community studying downstream processes. The presented results and conclusions are well presented, referenced and documented, with adequate descriptions of used methodologies, assumptions and uncertainty ranges. My assessment of the paper is that it is a novel, well-written and scientifically sound piece of work that is suitable for publication within the scope of Nature Communications.

We are buoyed by the Reviewer's positive perspective on our work, and appreciate the recognition of the novelty of the work, its appropriateness for Nature Communications, and the appropriate level of detail, referencing and rigour. We note that, in response to Reviewer 3, the biomass values cited here have been revised through a new upscaling approach and now are presented in mass per area units; we have removed the more speculative assessment, as requested by Reviewer 3.

However, before publication, I recommend that the manuscript is revised taking the following two minor issues into consideration:

1) The title of the paper does not imply that the study covers only a part of the Greenland Ice Sheet, which spans several climatic regions, with likely differences in the meltwater regime. Also, the title does not include the work on estimating the in-situ production of microbial carbon in the weathering crust which forms an essential part of the study. I recommend that the title is updated to more precisely reflect the area of the study domain (i.e. Western part of the GrIS) and that production is also included. A suggestion for a revised title could be "Production, storage and export of cellular carbon from the Western part of the Greenland Ice Sheet"

We agree with the Reviewer that our original title perhaps overplayed the study area's wider relevance, and so we have inserted "western" into the title to reflect this (now 12 words). However, owing to the fact the "production" of cellular carbon is based solely on theoretical evaluations using published literature (see Reviewer 3's comments), we feel the suggested title in full (16 words) perhaps misrepresents the data and work within the paper.

2) The abstract states that the authors estimate that up to 10³ tonnes of cellular carbon is

exported from the surface. This estimate is equal to the upper limit of the speculative guess (as mentioned above) and does not reflect the rather large uncertainty range that the data from the study suggest (i.e. from 130 kg to 950 ton carbon). I recommend that the wording of the abstract is rephrased to include this important uncertainty range, which also provides a clear argument for the need for further studies to constrain this uncertainty.

Again, we agree with Reviewer 2, that some subtle rephrasing of the last sentence of the Abstract would be appropriate to reflect the range rather than maximum value. We have made revisions here, but also through taking into account Reviewer 3's observations, we have revised our calculations and updated the values we quote in the Abstract. The Abstract, noting the range of values we derive, now closes with:

"We estimate that up to 37 kg km⁻² of cellular carbon is flushed from the near-surface environment of the western Greenland Ice Sheet each summer, providing a viable flux to support heterotrophs and methanogenesis at the bed."

Reviewer #3:

Please note that my expertise is in ice sheet hydrology, rather than flow cytometry enumeration and biogeochemistry.

This is an interesting and well-written paper that presents several noteworthy results.

We are grateful for the positive perspective the Reviewer reports for our manuscript.

Firstly, that accumulation of cellular carbon in the near-surface photic weathering crust exceeds fluvial export for the study's supraglacial catchment of the Greenland Ice Sheet. This is based on typical inter-stream distances, the speed of water flow in the weathering crust, and typical doubling times from other supraglacial environments. This near-surface accumulation of carbon would have important implications for the biological impact on the ice surface albedo (so-called 'bio-albedo'). This finding would, however, be much more robust if it were based on measurements of carbon in the weathering crust over the full 70-day melt season, rather than hydraulic conductivity and previously published cell accumulation rates.

We agree, as is always the case, more data is invaluable. However, we can not retrospectively create more data, and would point out this is a novel study (as both Reviewers 1 and 2 have highlighted); we seek to present the first estimate of carbon fluxes in such a logistically challenging environment. The focus remains on the daily time scale, with only a short section seeking to present the potential biomass values exported from the weathering crust in more regional and seasonal contexts. While we agree carbon within the weathering crust itself is an important characteristic to quantify, as our original discussion detailed, there are a range of components and causes of change therein through biogeochemical and hydrological processes; to assess these would require a very different and more integrated approach – which is what our paper argues for as future work. Our comments relating to 'bio-albedo' are indicative, and our data are not used to extend beyond the inference that accumulation of carbon in near surface ice may have an impact on its albedo. More detailed examination of microbial bio-albedo effects are available in papers such as Cook et al (2020: The Cryosphere) and Williamson et al (2020: PNAS) which are cited, and we therefore do not inappropriately extend beyond our own data here.

Secondly, the authors extrapolate their carbon flux data to the ablation area of the western Greenland Ice Sheet margin over an estimated 70-day bare-ice melt season. These up-scaled results indicate that between 400 and 950 tonnes of carbon is delivered to the ice sheet bed, and the majority of this reaches downstream ecosystems. Note that later (at the end of page 10) the authors mention that "seasonal biomass export to moulin-terminating supraglacial streams is 22 - 77 tonnes of cellular carbon", but it is not immediately clear to what temporal and spatial scales these numbers refer.

We recognise the Reviewer's point regarding clarity here, but note the 'best estimate' of 22-77 represents the 'allometric' biomass calculation, while the greater values represent use of the 'constant ratio' biomass calculation and inclusion of the 'large algae' class. We declared

the 'best estimate' clearly in our Methods section. These had been detailed, for the study catchment, in our original Table 1. We wonder if the Reviewer had not fully understood the various biomass conversion approaches we employed, or the caution we offered with regards inclusion of the larger algal class? Here, we note Reviewer 1 and 2 were very positive about our open presentation of the uncertainty surrounding the values we present. Nonetheless, we have completely revised our upscaling both over time (for the season) and for the wider western Greenland region (see details below), and this specific section text has now been entirely reworked and revised.

The field methods employed seem robust and the resulting data of high quality (in terms of ice sheet hydrology and DEM generation). My main (and I think a critical) issue is the extrapolation of data from a single small supraglacial catchment over 6 days, to a large part of the western margin of the Greenland Ice Sheet over a 70-day melt season. The upscaling method is not fully or clearly explained (there is just a reference to Yang and Smith 2016). In their extrapolation, how do the authors account for temporal and spatial variations in (amongst other things): ice albedo, ice surface melt rate, water table height, inter-stream distance, distribution of moulins and crevasses (which affect the proportion of a catchment that consists of water flow within the near-surface weathering crust compared to in supraglacial streams and rivers)? For such a crucial component of the main findings of the paper (and the result that would appeal to a broad audience), there needs to be a more robust and thorough presentation of the approach utilised.

In response to the Reviewer's observation that our upscaling approach was perhaps overly simplified, we have addressed this by completely revising our approach to the estimation for a border area over western Greenland. While the reviewer is correct in noting the variability of the weathering crust geometry, and potential drivers of that, there is almost no data available on the weathering crust, its spatial or temporal evolution. The data that does exist is limited to discrete study sites. Because the weathering crust forms as a result of both subsurface shortwave radiation receipt and heat fluxes from percolating meltwater, modelling the evolution of the weathering crust and any dependence on albedo or melt rate is not trivial. Models that, at least in part, seek to address this (e.g. Hoffman et al. 2014, J Glaciology) do exist. However, there is an absence of available data to constrain such models given the topic area we report is very much a burgeoning aspect of glacier hydrology, and in-depth modelling explorations would completely change the direction of this paper and are beyond its scope. We used the catchments described by Yang and Smith (2016, JGR), which are available in their open access data repository, because they are relatively free of crevasses, and terminate in moulins assumed to reach the ice sheet bed, and cover a region in western Greenland within which our study site lies. We recognise that the water table affects K , but at the same time, as K increases the microbe flux decreases; therefore, we argue using the mean value for K and cell abundance provides the best estimate of transported biomass. The reviewer is correct, in that the drainage density is a critical factor; however, this is defined by the resolution of the observational data, and studies using coarser elevation data (e.g. Arctic DEM at 5 m horizontal resolution) may fail to reveal the active streams at finer scales (e.g. 1 m).

However, as the Reviewer themselves notes, the upscaling is a critical component here. We therefore trust the entirely revised section “Microbial export from the weathering crust in western Greenland” and the new, additional “Supplementary Information” in support of the revised ‘upscaling’ provides an adequate response in terms of a more transparent methodology that incorporates some sense of a temporally evolving weathering crust and accommodates spatial variation to reflect the contrasts in bare ice duration over the region in western Greenland.

Specific points (by page number, P – note that it really would be useful if Nature Communications submissions came with line numbers).

Line numbers have been added to the revision for clarity and for future reference.

P1. (abstract) the authors use a combination of their measurements of hydraulic conductivity in the near-surface weathering crust along with previously published estimates of microbial productivity. I would argue that they therefore do not really ‘demonstrate that cellular carbon accumulation in the weathering crust exceeds fluvial export’, but rather they infer this.

We have changed “demonstrated” to “infer” to address the Reviewer’s concern over wording used here. We note this was also appropriate given our initial repetition of “demonstrate” in sequential sentences.

P1. (abstract) The last sentence of the abstract is really very speculative and relates back to my main criticism of the unexplained extrapolation of results from a single catchment over 6 days to a large part of the ablation area of the western margin of the ice sheet over an estimated 70-day melt season.

We note, as in our response to Reviewer 2, that there is indeed a need to be more circumspect here at the closing of the Abstract, through declaring the range of values not just the maximum. As noted above we have made substantial changes through a revised ‘upscaling’ of our results, and have accordingly updated the closing sentence of the Abstract (see response to Reviewer 2). Moreover, we would like to note that similar speculative estimates have been presented in numerous papers published by the Nature group, for example by Hawkings et al. (2014, Nature Communications; 2017 Nature Communications).

P6. “...is hydraulically-active at the ablating margin of the GrIS” is too broad based on the data presented. “...is hydraulically-active at our study catchment 38 km from the ablating margin of the GrIS” would be more accurate.

We have revised the text to clarify the study catchment’s position in western Greenland, and aligned this revision with a similar observation regarding the specific locality by Reviewer 2. Note, the revised title specifying “western Greenland” also aids in clarifying the study area.

P6. Might “using” be better than “under”?

Typographical error has been corrected. We appreciate the identification of the inaccurate wording.

P6. Do the authors know what proportion of the inter-stream area in the rest of the ablation area is dominated by the slow water flow through the weathering crust? It is entirely possible that at lower elevations particularly, most of the inter-stream water flow is through rills and micro-channels.

The Reviewer raises an interesting point here. Our study focuses on the uncrevassed, low-gradient portion of the western sector of the Greenland Ice Sheet (as described by Yang and Smith, 2016, JGR). This region is limited to elevations > 400 m a.s.l.. As such our upscaling does not specifically target the higher gradient (and more crevassed) regions found closer to the ice margin. Our focus is on supraglacial catchments that terminate in moulins. We do, however, note that the weathering crust is a ubiquitous phenomenon of ablating glacier surfaces, and while rills and micro-channels may well be important, neither our research, or other published work, has yet fully explored aspects such as the proportions of water flow through the complex medium that the weathering crust represents. However, the work by Muller and Keeler (1969) on an Arctic valley glacier does indicate that 1.3 cm of subsurface meltwater equivalent (or up to ~10% of surface ablation) over a 12 hr period may be released as runoff. The proportion of the total runoff derived from the saturated versus the unsaturated zones remains unconstrained, and here the latter is likely to be appreciably greater. Therefore, our revised text makes clearer reference to flow within the unsaturated weathering crust (which includes perched micro-channels) and notes that our study focuses on "... biomass efflux from the saturated weathering crust, ... and, therefore, represents a baseline".

P6. It seems likely that at times in the study catchment, and perhaps more frequently lower in the ablation area, the water table would rise up so that flow was principally through the higher porosity unsaturated ice layer. This would mean that the authors' estimates of nutrient and carbon storage are also time- and space-dependent and cannot be simplistically scaled up to the entire western margin of the ice sheet over a whole melt season.

As noted above, and in our results within the manuscript, as well as in the cited paper by Stevens et al (2018, Hydrological Processes), our data suggest the K value increases with a rising water table. We have unpublished data that suggests water table change may be of the order of 4 cm over a 24-hr period. However, as K increases, our data show the microbe abundance decreases. We therefore argue that using the average K value and cell abundance recorded over a 3 day period of observations provides a robust estimate, in the absence of a much more detailed assessment over time, which itself is logistically challenging given the recharge rate in near-surface ice. At no stage do we present a seasonal carbon storage value, nor any associated nutrient values. We are careful to keep these estimates to a daily time-scale that reflects the duration of our observation periods. We feel an expectation to have resolved all the uncertainties here is a little misplaced given the

novelty of the topic we are seeking to promote further study in, and given the clear declaration of the wide variety of uncertainties that we do indeed present. Our aim here is to provide a paper that serves as a platform for further work on the weathering crust and the transport of biomass through the interfluvial area; in so doing, we hope our paper would be highly cited as a foundational study.

P8. The high bound of the authors' estimate of carbon export (950 tonnes) seems over speculative – no quantification of the “heightened mean abundance reflective of the dust and algal blooms commonly found across the ice-sheet's south-western margin” is put forward.

We disagree with the Reviewer's comment here and would refer them to our original and now revised texts. Throughout our manuscript we were and have been careful to fully detail the range of values used or presented, and their justification. This transparency in the range of values was commended by Reviewer 1. In our original text, on the preceding page (p7), we noted:

“[i]f we consider the inclusion of elevated microbe abundance related to discrete dust concentrations or algal blooms within our sample set (see Methods), the mean abundance is increased to 6.12×10^4 cells mL⁻¹, and our stock rises to 3.9×10^9 cells m⁻² or 0.05 to 34.1 kg C km⁻².”

As such, we felt it was clear to what the “heightened mean abundance” referred to. Consequently, the derivation of the uppermost values is based on our data, data which had been presented and explained, and not some speculative ‘guess’.

However, we note, through our entirely revised upscaling section, we have amended the text that quoted these values and which Reviewer 3 suggested were potentially problematic.

P10. The “Implications...” section rests on the unjustified assumption that the study catchment is “broadly representative of the region's supraglacial environment”. This is quite an assumption and is unlikely to be justified given known spatial variations in (amongst other things): ice albedo, ice surface melt rate, water table height, inter-stream distance, distribution of moulins and crevasses. No evidence is presented that lends support to this assertion.

With respect, we completely disagree with the Reviewer 3's assertion here. The “Implications for the carbon cycle of the Greenland ice sheet” section, relied nearly entirely on the data from the sub-catchment studied, not from the upscaling. Our original text presented the carbon storage in kg C km², and discussed some uncertainties relating to the data we observed; we then used an upscaled estimate to merely emphasise the relevance of potential implications of the flushing of microbes and carbon to downstream environments; subsequently, we used the carbon budget on a daily basis as derived from our study catchment to explore potential carbon pathways which remain poorly constrained. We, therefore, find the Reviewer 3's comments on this section themselves rather unjustified, as it would seem the Reviewer made assumptions of the content (i.e. hydrology) based on the

opening two sentences. We have therefore removed the text that seems to have been misleading, and we have revised any note of the regional estimates according to our new upscaling approach.

With regard the specifics added here, Reviewer 3 raises some valid points, and some of which we have commented on above. However, to date, it is unclear how albedo affects the weathering crust development, particularly as the ‘rotting’ of the weathering crust ice while initiated by subsurface melting (which is related to albedo) is then maintained or extended by the percolation of meltwater not irradiance. As noted above, our own studies (Stevens et al., 2018, Hydrological Processes) found relationships between melt rates, water table height, and hydraulic conductivity to be ambiguous. While water table height does appear to relate to the hydraulic conductivity, we have shown here that the microbe fluxes decrease as this conductivity increases; therefore, simplistically, as water tables rise, hydraulic conductivity also rises but cell transport decreases, and vice versa, suggesting that over time the average conductivity and cell abundance would be sensible as estimates for deriving the cellular carbon flux. Inter-stream distance will, in the absence of higher-resolution drone imagery (as used here), be a function of the resolution of the data-sets used and will evolve as the drainage density changes over the season. We have endeavoured to use an informative snapshot to reveal microbe transport rates within the saturated weathering crust, and our now evolving model of the weathering crust used for the regional upscaling at least in part accommodates this drainage density related uncertainty.

Unfortunately, we do not have the answers to all of these intertwining process-based queries Reviewer 3 raises, as this is an area in glacial hydrology that has until now been completely overlooked. By introducing this novel area of glacier hydrological research through emphasising its biogeochemical relevance, we hope our paper will serve to prompt further work on these enduring, non-trivial research questions. We reiterate our revised text emphasises that: our study provides “a benchmark that invites future refinement through better constraining the hydrological configuration, dynamics and functioning of the supraglacial interfluvial environment”.

P11. “phenomena” should be “phenomenon” I think.

Typographic error has been corrected; “phenomenon” is correct as the usage is singular. We thank the Reviewer for the observation.

P11. “in the top few centimetres of the ice sheet surface” should be “in the top few centimetres of the ice sheet” I think.

“Surface” has been deleted for clarity.

P18. “...define THE supraglacial network...”

“The” has been inserted. We appreciate the identification of an omitted word.

P18. There is an order of magnitude range in the estimates of stream spacing, which is key to quantifying the relative importance of slow water flow in the near-surface weathering

layer (and thus the degree to which cells and carbon accumulate in this layer). It would be good to see a sensitivity analysis of the carbon storage and export estimations when varying the stream spacing during the spatial and temporal extrapolation.

We have revised the entire “Catchment upscaling” section (see above). However, please note our original manuscript critically commented on the imagery resolution (Weathering crust hydrology – p6) and hydrological model sensitivity (Supraglacial catchment characterisation – p18). Our paper is based on primary data, and uses higher resolution imagery than many equivalent ice sheet supraglacial hydrology papers using remote sensing products: quite clearly, using coarser resolution data (which may mask ‘real’ drainage density) will yield a greater stream spacing, and this will (i) reduce the cell export by reductions in the stream bank area fed by the weathering crust and (ii) increase carbon storage through greater non-stream area. As such, the imbalance between export and accumulation will be more exaggerated. Therefore, we are not clear what this ‘finding’ would add to the paper in any meaningful way. Moreover, with knowledge that supraglacial drainage density changes over the season or as a consequence of melt intensity, we suggest the “sensitivity analysis” requested here is a more appropriate as an in-depth modelling experiment which is beyond the scope of this paper and the data therein. We reiterate our concluding phrase that closes the manuscript: “there remains a need to better constrain microbial carbon cycling pathways and their controls across supraglacial... environments in Greenland”.

P20. (Fig. 1 caption) “perceptive” should be “perspective” I think. While the perspective approach does allow the authors to show all three mapped layers in a small space, a more traditional planar display of the data would be clearer and show them in their entirety.

We have corrected the typographical error; we appreciate the observation. However, we disagree with the Reviewer regarding the detail of the Figure 1. The catchment has an approximately 1:10 width-length ratio, and presenting three planimetric maps we feel would neither present the data sufficiently clearly nor provide the reader with any further insight that is not afforded by the perspective plot. Moreover, the data is presented in an illustrative manner, given the paper’s focus is the recharge data and flow cytometric enumeration.

REVIEWERS' COMMENTS

Reviewer #2 (Remarks to the Author):

Thank you for the opportunity to review the revised manuscript. I am satisfied with the improvements to the original manuscript that is now presented, and have no further comments.

Reviewer #3 (Remarks to the Author):

The authors have thoughtfully and thoroughly addressed my comments on the original manuscript. I acknowledge that I did misinterpret part of the discussion, but hopefully by doing so, perhaps also highlighted where the manuscript could be clarified for a broader audience. The new section which details the upscaling approach is a welcome addition and the revised abstract also better reflects the findings of the paper. I believe that the revised manuscript is now suitable for publication within Nature Communications.

Response to the Editor

Editor comments in black standard text, our responses indicated in italic red text.

Your manuscript entitled "Storage and export of microbial biomass across the western Greenland Ice Sheet" has now been seen again by our referees, whose comments appear below. In light of their advice I am delighted to say that we are happy, in principle, to publish a suitably revised version in Nature Communications under the open access CC BY license (Creative Commons Attribution 4.0 International License).

We therefore invite you to revise your paper one last time to address the remaining concerns of our reviewers and our editorial requests in the attached document(s). At the same time we ask that you edit your manuscript to comply with our policies and formatting requirements and to maximise the accessibility and therefore the impact of your work.

We are very pleased that our revisions, following helpful commentary from three Reviewers, has been viewed as positively as this. We have followed the variety of points of feedback that the Editorial Office has offered: specifically, we have revisited the manuscript and we believe this now meets the requirements of Nature Communications in terms of titles and subtitles, formatting, figure design, and referencing style. We have, as detailed in our cover letter and our Author Checklist document, made a number of very minor changes – these include formatting, word choice, and splitting of our original three-part Figure 4 into Figures 4, 5a and 5b. Most critically we have, now no longer being in the DBPR phase, updated all the back-matter, particularly the Data Availability which now includes a number of DOIs for open-access repositories. Please refer to our cover letter and detailed responses in the Author Checklist for full information.

Note, we have also, in response to feedback on the Summary Report and Policy documents, updated, rectified and revised these in accordance with the data, funders and Nature's ethics and policies.

We hope that the response to the editorial requests now ensures our manuscript is in a position to move forward with publication.

Response to Reviewer comments

Reviewer comments in black standard text, our responses indicated in italic red text.

Reviewer #2:

Thank you for the opportunity to review the revised manuscript. I am satisfied with the improvements to the original manuscript that is now presented, and have no further comments.

We are pleased to learn our revisions on the earlier version of the manuscript have been viewed as positively.

Reviewer #3:

The authors have thoughtfully and thoroughly addressed my comments on the original manuscript. I acknowledge that I did misinterpret part of the discussion, but hopefully by doing so, perhaps also highlighted where the manuscript could be clarified for a broader audience. The new section which details the upscaling approach is a welcome addition and the revised abstract also better reflects the findings of the paper. I believe that the revised manuscript is now suitable for publication within Nature Communications.

We are very grateful to the Reviewer for indeed highlighting some aspects in the earlier version of the manuscript that could be clarified to readers. Similarly, we are pleased to learn that our revised upscaling approach, detailed in our new Supplementary Information documentation proved to be such a welcome advance on our findings. We thank the Reviewer for positive and constructive commentary and that careful reworking of several sections in the paper have been and improvement.